# A ChatGPT Aided Explainable Framework for Zero-Shot Medical Image Diagnosis

Jiaxiang Liu [* 1 2]   Tianxiang Hu [* 1]   Yan Zhang [* 3]   Xiaotang Gai [1]   Yang Feng [4]   Zuozhu Liu [1]

## Abstract

Zero-shot medical image classification is a critical process in real-world scenarios where we have limited access to all possible diseases or large-scale annotated data. It involves computing similarity scores between a query medical image and possible disease categories to determine the diagnostic result. Recent advances in pretrained vision-language models (VLMs) such as CLIP have shown great performance for zero-shot natural image recognition and exhibit benefits in medical applications. However, an explainable zero-shot medical image recognition framework with promising performance is yet under development. In this paper, we propose a novel CLIP-based zero-shot medical image classification framework supplemented with ChatGPT for explainable diagnosis, mimicking the diagnostic process performed by human experts. The key idea is to query large language models (LLMs) with category names to automatically generate additional cues and knowledge, such as disease symptoms or descriptions other than a single category name, to help provide more accurate and explainable diagnosis in CLIP. We further design specific prompts to enhance the quality of generated texts by Chat-GPT that describe visual medical features. Extensive results on one private dataset and four public datasets along with detailed analysis demonstrate the effectiveness and explainability of our training-free zero-shot diagnosis pipeline, corroborating the great potential of VLMs and LLMs for medical applications.

---

[*]Equal contribution   [1]Zhejiang University-University of Illinois at Urbana-Champaign Institute, Zhejiang University, Haining, China [2]College of Computer Science and Technology, Zhejiang University, Hangzhou, China [3]National University of Singapore, Singapore [4]Angelalign Inc., Shanghai, China. Correspondence to: Zuozhu Liu <zuozhuliu@intl.zju.edu.cn>.

*Workshop on Interpretable ML in Healthcare at International Conference on Machine Learning (ICML)*, Honolulu, Hawaii, USA. 2023. Copyright 2023 by the author(s).

## 1. Introduction

Large-scale pretrained vision-language models (VLMs), such as the Contrastive Language-Image Pre-Training (CLIP), have shown great performance in various visual and language tasks, especially in zero-shot recognition tasks (Radford et al., 2021; Li et al., 2022). In the standard zero-shot image classification scenario, CLIP computes similarity scores between a query image and different category names (texts), and the category with the highest similarity score would be regarded as the classification result (Menon & Vondrick, 2022). Recent work extends ideas in CLIP to medical image analysis, e.g., how CLIP benefits medical image classification or the training of large-scale VLMs such as MedCLIP in the medical domain (Eslami et al., 2021; Shen et al., 2021; Wang et al., 2022).

One of the key tasks to extend VLMs in medical domains is zero-shot medical image classification, which is critical in real-world scenarios where we may not have access to all possible diseases or annotated medical images are hardly available (Mahapatra et al., 2021; 2022). However, this significant task remains seldomly explored especially with VLMs, let alone frameworks for explainable diagnosis with visual or textual medical information. The feasibility of directly transferring VLMs like CLIP to medical domains remains to be checked for at least two reasons. On one hand, CLIP and many other VLMs are pre-trained with natural image-text pairs, which could certainly lack the attention on medical information and lead to abysmal performance (Chen et al., 2019; 2020; Li et al., 2020). On the other hand, medical image classification does embrace model interpretability, while the category texts of medical images tend to be highly abstract medical lexicons that are inherently more challenging to interpret and analyze with existing VLMs (Wang et al., 2022; Mahapatra et al., 2021).

It is reasonable to follow the standard zero-shot classification paradigm in natural image recognition for medical image classification (Menon & Vondrick, 2022). As illustrated in Figure 1 (top row), an input image, such as a brain MR or fundus image, and the possible categories could be fed into the multimodal CLIP model to compute corresponding similarity scores and subsequently make decisions. However, such a standard pipeline suffers from the afore-

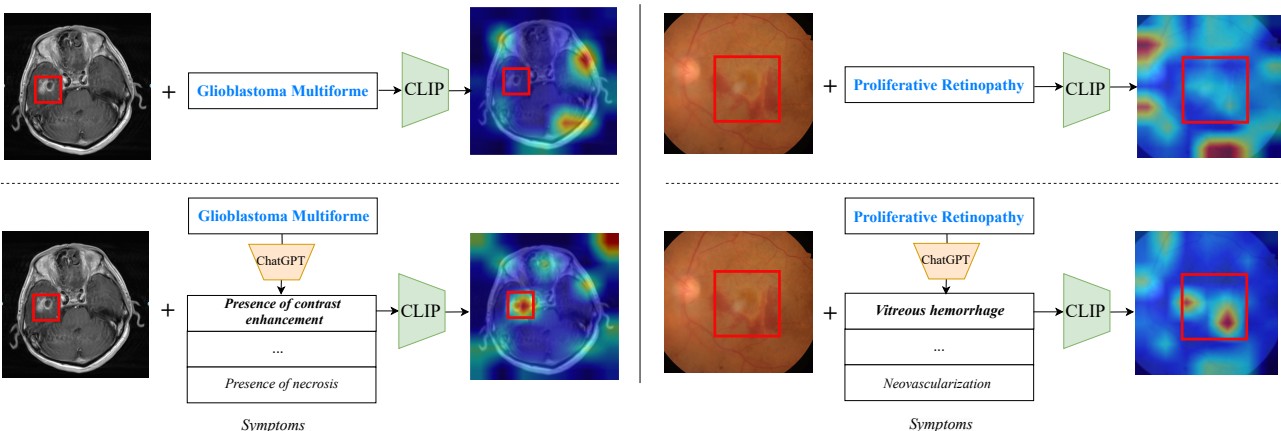

*Figure 1.* Attention maps of query image with only diagnostic category or with additional symptoms from ChatGPT, generated with model CLIP-ViT-B/32 which is used for all attention maps in the rest of the paper (Chefer et al., 2021).

mentioned limitations with inferior performance and limited interpretability, i.e., the attention map generated by the representations of the image and category name show that the model fails to focus on the area of interest for identifying the diagnostic category (Chefer et al., 2021). A simple idea is providing additional information about each disease to assist diagnosis, and if possible, providing them in a scalable way to avoid time-consuming hand-crafting. For instance, one may consider utilizing generative models to create the descriptions, with which CLIP can inference based on certain symptoms, very much like the diagnostic process performed by human experts in practice.

In this paper, we propose a novel framework for explainable zero-shot medical image classification. The key idea is to leverage LLMs to automatically generate additional cues and knowledge, such as disease symptoms or descriptions other than the standalone category name, to help provide more accurate and explainable diagnosis. The effectiveness of our method is illustrated in Figure 1 (bottom row), where the model obviously pays more attention to relevant tissues after incorporating the information of more detailed descriptions. In particular, besides leveraging VLMs like CLIP, we supply our model with the recently released ChatGPT model to automatically provide detailed category descriptions. Considering that ChatGPT may deliver inaccurate information in medical queries, we further design a prompt to query ChatGPT to get textual descriptions of useful visual symptoms to identify diagnostic categories. Extensive experimental results demonstrate the superiority of our method. The main contributions could be summarized as:

- We have shown for the first time in the medical domain the feasibility of incorporating ChatGPT for better CLIP-based zero-shot image classification, reveal-

ing the potential of LLMs-aided designs for medical applications. Through ablation studies, we have also elucidated the considerable scope for augmenting the performance of our approach through prompt designs.

- We propose a novel CLIP-based zero-shot medical image diagnosis paradigm. In comparison to the conventional CLIP-based approach, our proposed paradigm exhibits significant enhancements in medical image classification accuracy while concurrently offering a notable level of explainability in various disease diagnosis.

- We comprehensively evaluate our method on five medical datasets, including pneumonia, tuberculosis, retinopathy, and brain tumor. Extensive experiments and analysis demonstrate the promising zero-shot recognition performance and a considerable level of interpretability of our approach.

## 2. Related Work

### 2.1. Large Language Model

Large language models use deep neural networks with billions of parameters to learn patterns from large amounts of text data (Brown et al., 2020; Devlin et al., 2018; Radford et al., 2019), the primary objective of which is to generate human-like responses based on the context it is provided (Ouyang et al., 2022). ChatGPT is a state-of-the-art large language model that specializes in language understanding and inference. With the advanced transformer architecture and massive training dataset, ChatGPT is able to deliver coherent and meaningful responses to a wide variety of prompts including questions consisting of a few words and

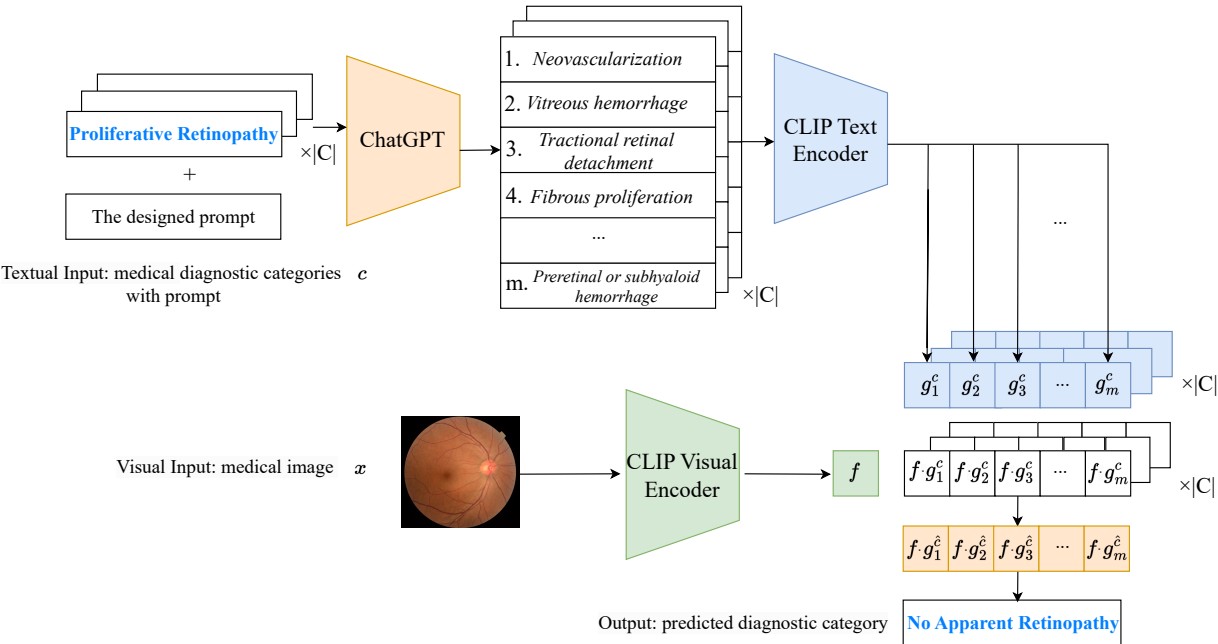

*Figure 2.* The pipeline of the proposed method.

complex dialogues. It has shown an impressive performance in many language tasks such as machine translation, text summarization, conversation generation, and sentiment analysis (Houlsby et al., 2019; Karimi Mahabadi et al., 2021; Mahabadi et al., 2021). News popped up that ChatGPT earned a decent grade on the US Medical Licensing Exam, but the well noticed fact that ChatGPT can generate make-up knowledge have raised a dispute over the issue of whether ChatGPT can be used in medical diagnosis. Our proposed approach introduces an innovative and prospective paradigm that can integrate LLMs like ChatGPT into medical diagnosis, yielding promising performance in explainable zero-shot medical image diagnosis.

### 2.2. Vision-language Pre-training

Visual-language (VL) pre-training is meant to pre-train multi-modal models on large-scale datasets that contain both visual and textual information (Chen et al., 2019; Do et al., 2021; Liu et al., 2021), eg. images and captions, to learn joint representations that capture the complex interactions between the two modalities. In practice, due to the high cost of acquiring manually annotated datasets, most visual-language models (Chen et al., 2019; 2020; Li et al., 2020; Radford et al., 2021) are trained with image-text pairs captured from the Internet (Jia et al., 2021; Sharma et al., 2018). As an important example, with pre-training on 400 million pairs of image and text from the Internet, the model

CLIP (Radford et al., 2021) gained rich cross-modal representations and achieved amazing results on a wide range of visual tasks without any fine-tuning. Based on the abundant knowledge CLIP has learned, we are able to establish the framework to tackle medical image diagnosis tasks in a training-free manner.

## 3. Method

### 3.1. Problem Formulation and Method Pipeline

We focus on zero-shot medical image classification tasks where we compute the similarity scores for image-text query pairs $(x, c), c \in C$, and the category $\hat{c}$ with highest similarity score would be regarded as the classification result of image $x$, where $C$ is the label set. Note that there is no need for model training as long as we have pretrained VLMs or LLMs in our framework. The pipeline is illustrated in Figure 2. The image $x$ is processed through the visual encoder of CLIP to obtain its visual representation:

$$f = \text{VisualEncoder}(x). \quad (1)$$

In parallel, ChatGPT is queried with our designed prompt to generate major symptoms for each diagnostic category:

$$s_1^c, ..., s_m^c = \text{ChatGPT}(prompt, c), \quad (2)$$

where $m$ denotes the total number of generated symptoms, see details in next section. The symptom phrases are then

sent into the text encoder of CLIP to obtain text representations:

$$g_1^c, ..., g_m^c = \text{TextEncoder}(s_1^c, ..., s_m^c). \qquad (3)$$

A score function $S$ is defined to evaluate the similarity of the image-text pair $(x, c)$ at the feature level:

$$S(x, c) = \frac{1}{m} \sum_{i=1}^{m} f \cdot g_i^c. \qquad (4)$$

We use average aggregation to ensure the fair evaluation for classes with different sizes of symptom, and more aggregation strategies are evaluated in experiments. A high score of $x$ and $c$ suggests a significant degree of relevance between the medical image and the class. Going over all categories $c \in C$, the one with the maximum similarity score is finally taken as the predicted diagnosis of the input image $x$:

$$\hat{c} = \underset{c \in C}{\text{argmax}}\, S(x, c) = \underset{c \in C}{\text{argmax}}\, \frac{1}{m} \sum_{i=1}^{m} f \cdot g_i^c. \qquad (5)$$

### 3.2. The Designed Prompt

High-quality symptom descriptions are essential for the success of our method. In this work, we query ChatGPT useful symptoms for the diagnosis of certain diseases. A baseline prompt choice could be "Q: What are useful visual features for distinguishing {Diagnostic Category} in a photo?". However, in experiments we found that such baseline prompts can produce misleading symptom descriptions in the returned answer. Hence, we secure the fidelity of the generated information by designing the prompts to work more professionally. First, noticing that the object of the method is basically to replicate the diagnostic process carried out by medical professionals, we emphasize that the generation should focus on medical features in the prompt instead of general descriptions as if we are querying explanations for the class of a general natural image. In addition, we adjust the prompt to direct ChatGPT's attention more toward published literature. Figure 3 exhibits the symptoms generated by ChatGPT using our designed prompt. As expected, generated symptoms center around the diagnostic category, which typically involves the presence or absence of certain structure, location and clarity of relevant tissues, descriptions of organ boundaries, etc.

## 4. Experiments

### 4.1. Datasets and Experimental Setup

#### 4.1.1. DATASETS

**Pneumonia Dataset:** The Pneumonia Chest X-ray Dataset consists of images selected from retrospective cohorts of pediatric patients of one to five years old from Guangzhou

Women and Children's Medical Center in Guangzhou, Guangdong Province, China (Kermany et al., 2018). A total of 5,232 chest X-ray images of children are collected and labeled, including 3,883 characterized as having pneumonia (2,538 bacterial and 1,345 viral) and 1,349 normal. **Montgomery Dataset:** The Montgomery County X-ray Set was made in the tuberculosis control program of the Department of Health and Human Services of Montgomery County, MD, USA (Jaeger et al., 2014). The dataset contains 138 posterior-anterior X-rays, of which 80 X-rays are normal and 58 X-rays are abnormal with manifestations of tuberculosis, covering a wide range of abnormalities including effusions and miliary patterns. **Shenzhen Dataset:** The Shenzhen Hospital X-ray Set was collected by Shenzhen No.3 Hospital in Shenzhen, Guangdong Province, China (Jaeger et al., 2014), where the X-rays are acquired as part of the routine care at the hospital. There are 326 normal X-rays and 336 abnormal X-rays showing various manifestations of tuberculosis. **IDRID Dataset:** The Indian Diabetic Retinopathy Image Dataset is the first database representative of an Indian population regarding Diabetic Retinopathy, and is the only dataset that constitutes typical diabetic retinopathy lesions (Porwal et al., 2018). The dataset has 516 rentinal images in total, and provides information on the disease severity of diabetic retinopathy(DR) for each image according to medical experts' grading with a variety of pathological conditions of DR. **BrainTumor Dataset:** All datasets mentioned previously are composed of publicly available images which may potentially be part of the resource for CLIP's pre-training. To check the capability of our method on data that is absolutely invisible for CLIP, we originally construct a dataset of brain tumor images, specifically targeting glioblastoma multiforme and primary central nervous system lymphoma which are often hard to discriminate, even for medical professionals. The dataset encompasses 338 glioblastoma multiforme images and 255 primary central nervous system lymphoma images.

#### 4.1.2. EXPERIMENTAL SETUP

All implementations are based on the PyTorch framework and on an Ubuntu server with a single NVIDIA GEFORCE RTX 3090 GPU. For visual and textual encoders, we utilize five CLIP versions including RN50, RN101, RN50x64, ViT-B/32, and ViT-L/14, covering different levels of parameter size.

### 4.2. Main Results

#### 4.2.1. CLASSIFICATION PERFORMANCE

We evaluate our method on the five datasets across different CLIP versions. Results are reported in Table 1 where the best accuracies are displayed in the bottom row. Results show that our approach can achieve consistent improve-

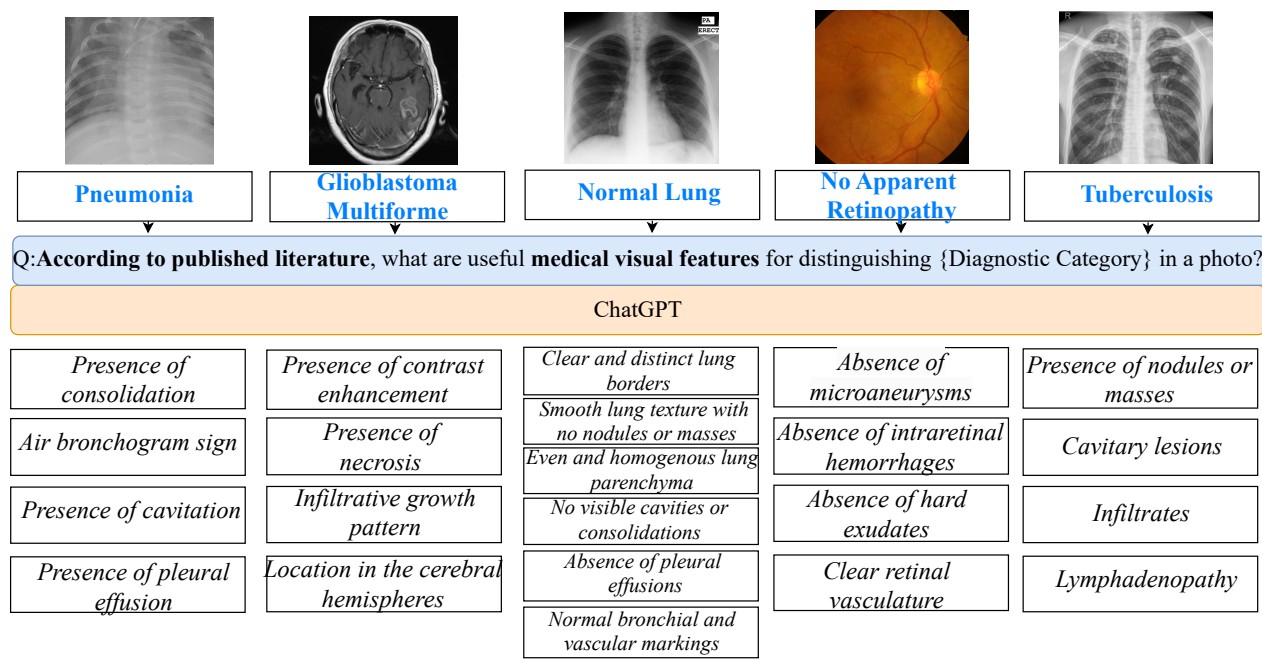

*Figure 3.* Visual symptom descriptions generated by ChatGPT with our designed prompt.

ments over the standard zero-shot classification using CLIP across all datasets, where the method has raised the zero-shot binary classification accuracy to over 62%, and the effectiveness of the method is especially evident on the Pneumonia dataset and Shenzhen dataset with improvements of up to 11.73% and 17.37% respectively. Remarkably, on the Shenzhen dataset, applying different CLIP versions without incorporating LLM yields a uniform accuracy of 50.76% due to the incapability of diagnosis which results in the misclassification such that all X-ray images are identified as abnormal (Shenzhen dataset includes 336 abnormal X-rays and 326 normal X-rays, $50.76\% \approx \frac{336}{336+326}$), yet our method can reach accuracy as high as 68.13%. Such findings indicate the presence of inherent potential within CLIP for undertaking medical image classification tasks such as identifying "Tuberculosis", where the potential remains largely untapped when CLIP is employed directly, and can be unleashed to a considerable extent with our designs.

### 4.2.2. INTERPRETABILITY

As shown in Figure 4, we conduct statistical analysis on selected medical images where our method successfully predicts the diagnostic category yet CLIP fails. For any such an image, specifically, we calculate the similarity degrees between the image and generated texts for both the true image class and the alternative. The pink bars in our visualizations indicate the similarity between the medical images and the text characteristics of the true category hinted by ChatGPT, while the green bars indicate the similarity between the medical images and the text (identified by ChatGPT) of the category inferenced by the CLIP. It is evident that the accuracy of our framework judgment is due to the high similarity between the images and the correct category characteristics. For instance, Figure 4 (a) shows that the characteristics of normal lungs exhibit higher similarity compared to those of tuberculosis in a comprehensive manner, as the majority of normal lung characteristics displays superiority. Our framework thereby identifies the image to be normal lungs according to prominent texts such as "No visible cavities or consolidations", "Absence of pleural effusions", "Clear and distinct lung borders". Figure 4 (b) shows that "Venous beading and loops" and "Neovascularization" exhibit the highest similarity among all characteristics, which leads to the identification of "Severe Nonproliferative Retinopathy". In the other two instances as illustrated in Figure 4 (c) and Figure 4 (d), dominant characteristics are observed. Specifically, Figure 4 (c) and Figure 4 (d) shows the conspicuous prominence of the similarity exhibited by "Air bronchogram sign" and "Restricted diffusion on MRI" which leads to successful identification of "Pneumonia" and "Primary Central Nervous System Lymphoma" respectively. Generally, a significant portion of generated visual symptoms for the true diagnosis possess a dominant similarity, indicating that our framework recognizes unique visual patterns to arrive at an accurate diagnosis.

*Table 1.* Accuracy (%) and gains (Ours *vs.* CLIP with only category names).

| | Pneumonia | | Montgomery | | Shenzhen | | BrainTumor | | IDRID | |
|---|---|---|---|---|---|---|---|---|---|---|
| | Ours | CLIP | Ours | CLIP | Ours | CLIP | Ours | CLIP | Ours | CLIP |
| RN50 | 76.28 | 27.41 | 58.70 | 57.97 | 50.91 | 50.76 | 51.95 | 53.47 | 25.24 | 13.59 |
| RN101 | 72.97 | 27.05 | 63.77 | 59.42 | 50.76 | 50.76 | 46.19 | 57.36 | 19.45 | 12.62 |
| RN50x64 | 71.26 | 53.42 | 57.97 | 57.97 | 58.61 | 50.76 | 57.02 | 57.02 | 07.77 | 21.36 |
| ViT-B/32 | 72.90 | 27.08 | 57.25 | 56.52 | 51.51 | 50.76 | 56.35 | 56.85 | 17.48 | 10.68 |
| ViT-L/14 | 73.36 | 64.55 | 59.42 | 60.14 | 68.13 | 50.76 | 62.61 | 57.36 | 20.38 | 06.80 |
| Best Acc | **76.28**$_{+11.73}$ | 64.55 | **63.77**$_{+3.63}$ | 60.14 | **68.13**$_{+17.37}$ | 50.76 | **62.61**$_{+5.25}$ | 57.36 | **25.24**$_{+3.88}$ | 21.36 |

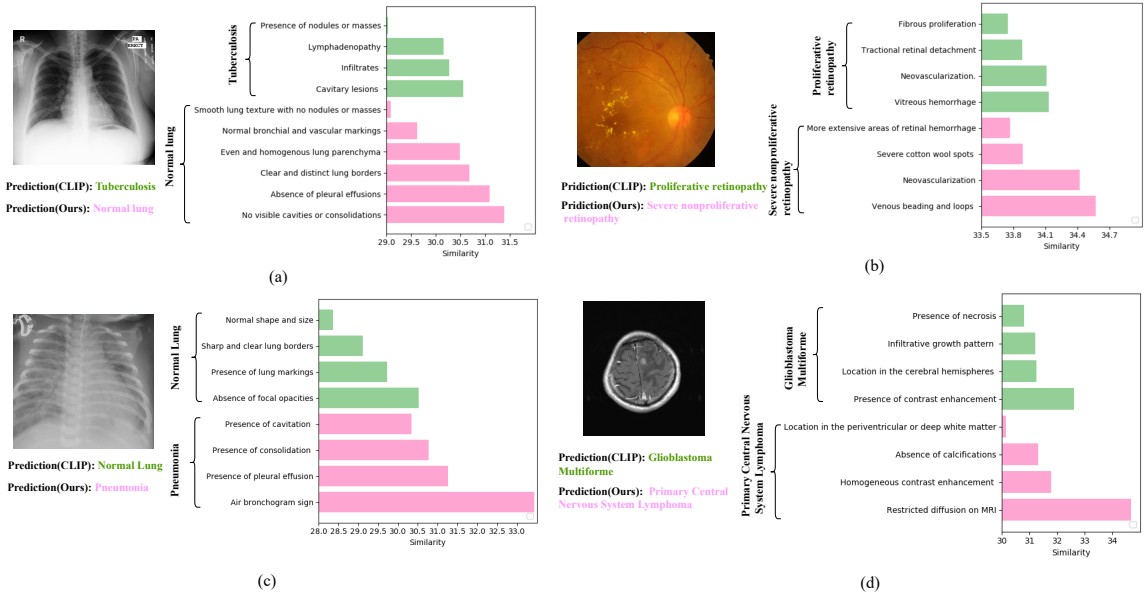

*Figure 4.* Explainability analysis with computed similarity scores on various symptoms.

### 4.2.3. CASE STUDY

Furthermore, we also provide two case studies with visualizations to see how the symptoms generated by ChatGPT contribute to classification decisions, as illustrated in Figure 5. In particular, we compare attention maps from our method with detailed symptoms to those produced by CLIP with only diagnostic category names. The first case exhibits proliferative retinopathy. We can observe that with the additional information provided by symptom texts such as "Fibrous proliferation", "Tractional retinal detachment", "Vitreous hemorrhage", the model's attention is increasingly drawn to the scar tissue on the retina. One notable example is "Tractional retinal detachment", which refers to the separation of the retina from the retinal pigment epithelium due to the pull of hyperplasia of fibrous tissue or scar tissue. In this case, ChatGPT generates the symptom text "Tractional retinal detachment", while CLIP focuses on the scar tissue in accordance with the text generated by ChatGPT, leading to the successful identification of "Proliferative Retinopathy". Likewise, the second case of primary central nervous system lymphoma indicates that symptoms such as "Absence of

calcifications" and "Homogeneous contrast enhancement" direct the model to concentrate more on the tumor area.

### 4.3. Ablation Study

#### 4.3.1. EFFECTIVENESS OF THE DESIGNED PROMPT

We compare the performance of the designed prompt "Q: According to published literature, what are useful medical visual features for distinguishing {Diagnostic Category} in a photo?" to the baseline prompt "Q: What are useful visual features for distinguishing {Diagnostic Category} in a photo?". Table 2 shows the superiority of ours on four out of five datasets, including the private one that CLIP has absolutely never encountered before. We attribute the advancement to preciser information acquired through a more appropriate querying, which is further demonstrated in Figure 6, where our prompt results in more attention around the upper lobes of lungs, the area of interest for identifying tuberculosis. Figure 6 also shows that the baseline prompt can generate noisy symptoms, which may confuse the diagnosis.

*Table 2.* Accuracy (%) of our framework with the designed and the baseline prompts.

| | Pneumonia | | Montgomery | | Shenzhen | | BrainTumor | | IDRID | |
|---|---|---|---|---|---|---|---|---|---|---|
| | DP | BP | DP | BP | DP | BP | DP | BP | DP | BP |
| RN50 | 76.28 | 73.00 | 58.70 | 59.42 | 50.91 | 53.17 | 51.95 | 55.84 | 25.24 | 23.30 |
| RN101 | 72.97 | 72.95 | 63.77 | 59.42 | 50.76 | 50.76 | 46.19 | 51.27 | 19.45 | 13.60 |
| RN50x64 | 71.26 | 75.03 | 57.97 | 57.97 | 58.61 | 49.85 | 57.02 | 57.02 | 07.77 | 33.01 |
| ViT-B/32 | 72.90 | 72.92 | 57.25 | 46.38 | 51.51 | 50.76 | 56.35 | 56.68 | 17.48 | 14.56 |
| ViT-L/14 | 73.36 | 72.97 | 59.42 | 63.77 | 68.13 | 64.65 | 62.61 | 58.04 | 20.38 | 18.45 |
| Best Acc | **76.28**$_{+1.25}$ | 75.03 | **63.77**$_{+0}$ | 63.77 | **68.13**$_{+3.48}$ | 64.65 | **62.61**$_{+4.57}$ | 58.04 | 25.24 | **33.01** |

DP denotes the designed prompt; BP denotes the baseline prompt.

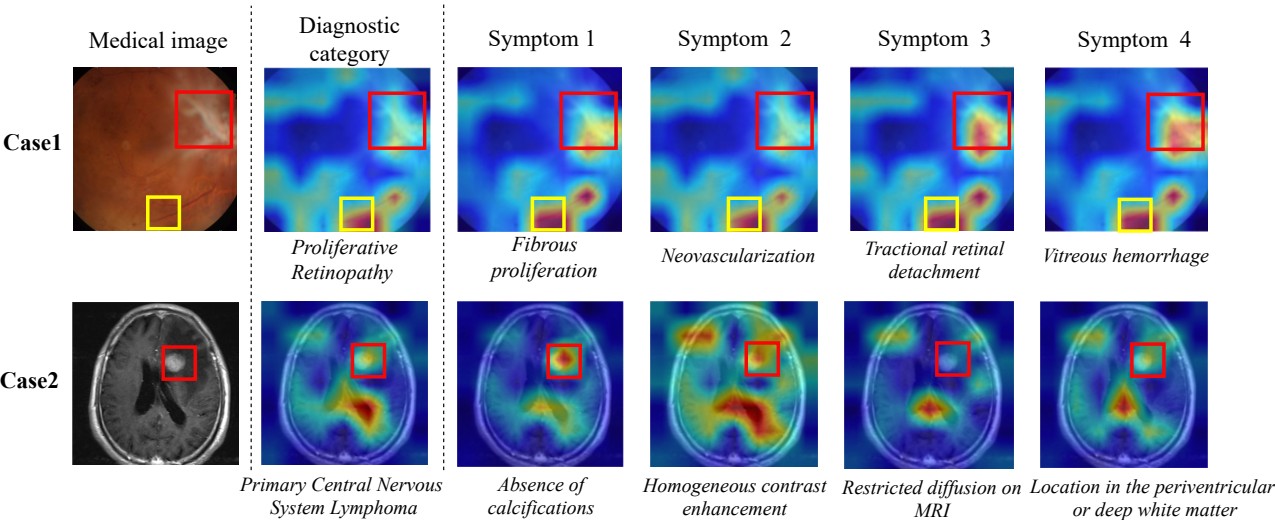

*Figure 5.* Attention maps generated by combining medical image with only diagnostic category or additional textual symptoms from ChatGPT.

### 4.3.2. EFFECTIVENESS OF DIFFERENT AGGREGATIONS STRATEGIES TO COMPUTE $S(x, c)$

For computing $S$, we consider operations *mean* and *max* to aggregate $f \cdot g_i^c$. Table 4 in appendix records prediction accuracies of each approach across all CLIP versions where the best results are displayed in the bottom row. Results indicate that aggregation operation *mean* performs best on almost all datasets.

### 4.4. Our framework *vs.* OpenFlamingo

We conduct a comparison between our framework and existing open source multimodal large models, such as Open-Flamingo (Awadalla et al., 2023). Flamingo (Alayrac et al., 2022)/OpenFlamingo (Awadalla et al., 2023) incorporates new gated cross-attention-dense layers within a frozen pretrained LLM to condition the LLM on visual inputs. The keys and values in these layers are derived from vision features, while queries are derived from language inputs. For

medical image diagnosis in OpenFlamingo, medical visual question answering is adopted, where the question is "Is this an image of {Diagnostic Category}?" In our experiments, we use OpenFlamingo 9B model, an open-source replica of the DeepMind Flamingo, trained on 5M samples from the new Multimodal C4 dataset (Zhu et al., 2023) and 10M samples from LAION-2B.

Experiments show that our framework outperforms Open-Flamingo in most datasets for medical image diagnosis. While OpenFlamingo achieves 100% accuracy for the Shenzhen dataset, it falls short in other datasets. The 100% accuracy may be attributed to the inclusion of the Shenzhen dataset within OpenFlamingo's pre-training dataset. In the other datasets, our method achieves a performance gain of 2.59% to 5.80% over OpenFlamingo for the zero-shot task. Notably, our interpretable framework surpasses the accuracy of OpenFlamingo by 5.59% in our constructed private dataset, BrainTumor. In summary, experimental results indicate that our interpretable framework can be superior to large

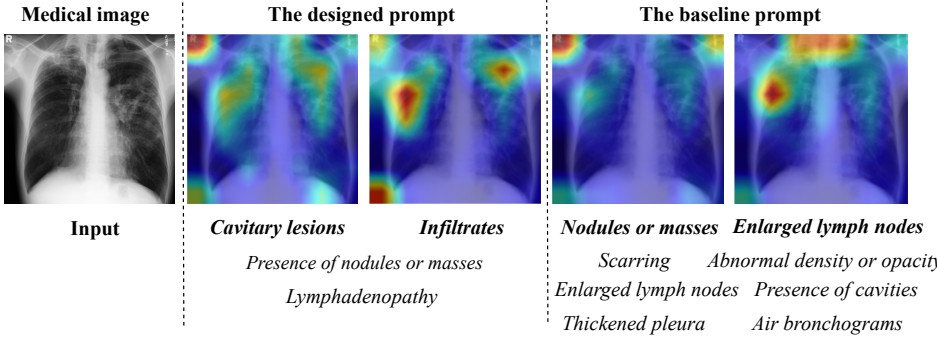

*Figure 6.* Attention maps with the designed and baseline prompts.

*Table 3.* Accuracy (%) comparison of our framework and Open-Flamingo.

|  | Ours | OpenFlamingo |
|---|---|---|
| Pneumonia (Kermany et al., 2018) | **76.28** | 72.97 |
| Montgomery (Jaeger et al., 2014) | **63.77** | 57.97 |
| Shenzhen (Jaeger et al., 2014) | 68.13 | **100** |
| BrainTumor | **62.61** | 57.02 |
| IDRID (Porwal et al., 2018) | **34.95** | 32.36 |

multimodal pre-training models such as OpenFlamingo in terms of medical diagnosis.

### 4.5. Discussion

Our work presents an initial trial of zero-shot medical image diagnosis with LLMs and VLMs. Our proposed paradigm can greatly unleash the power of VLMs (We use CLIP in our experiments) to provide explainability within the medical image diagnosis process and achieve noteworthy zero-shot medical image classification accuracy boost. Except for one dataset, using our method, the zero-shot image classification accuracy with CLIP has been raised to over 62%, and that on two datasets over five exhibits an improvement over 10%, which certainly increases the optimism of a decent diagnosis accuracy using such a low-cost approach that requires no extra network training at all, motivating investigation of more scenarios with VLMs and LLMs, such as data-efficient and few-shot learning in multimodal medical data analysis.

It is intriguing that with a slight modification of the prompt, the prediction accuracy is improved on almost all datasets, which indicates the potential of enhancement that can be brought by better querying, calling for more works concerning well-designed prompts. Another aspect to be discovered for upgrading is the multi-modal feature aggregation mechanism. We have explored the mean, max approaches in our experiments, in which mean performs the best. More strategies can be examined in future work. For example, instead of using mean, one may consider evaluating the significance

of different symptoms, perhaps with the help of ChatGPT, and create a more effective weight setting accordingly.

One shortcoming of our method is its unsatisfactory performance on the IDRID dataset, which could be a result of the inherent challenge in the recognition task. Rather than to detect the presence of certain diseases as in every other dataset, IDRID requires to evaluate the severity, in which the distinction space between classes is undoubtedly smaller. The difficulty has also been reflected in the sub-optimal accuracy values of supervised learning methods (Jang et al., 2022; Luo et al., 2020; Wu et al., 2020), which are reported very recently. In addition, the performance of zero-shot classification is still not on par with the supervised counterparts, as shown in Table 5 in Appendix. We envision this gap would soon narrow down with better designs of architectures and prompts.

Another limitation of our method is that it does not address the issue of hallucinations or inaccuracies in disease diagnosis that may be caused by ChatGPT in more complex medical scenarios. This is a significant concern in medical image recognition, and advanced algorithms must be developed to improve the accuracy of disease diagnosis provided by ChatGPT. While the prompt design proposed in this paper provides a solution, it is not a perfect one. We plan to further investigate this issue in the future to mitigate the problem of ChatGPT hallucinations in the medical field.

### 5. Conclusion

In this work, we propose an explainable framework for zero-shot medical image classification by integrating ChatGPT and CLIP. Extensive experiments on five challenging medical datasets, including pneumonia, tuberculosis, retinopathy, and brain tumor, demonstrate that our method is able to carry out explainable diagnosis and boost zero-shot image classification accuracy. We hope our work could encourage more in-depth research that leverages VLMs and LLMs for efficient, accurate and explainable medical diagnosis.

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

# A. Appendix

*Table 4.* Ablation on two aggregation operations (Best Acc$_{DP}$ and Best Acc$_{BP}$ denote the best accuracy (%) of the designed and baseline prompts.)

|  | Pneumonia | | Montgomery | | Shenzhen | | BrainDataset | | IDRID | |
|---|---|---|---|---|---|---|---|---|---|---|
|  | Mean | Max | Mean | Max | Mean | Max | Mean | Max | Mean | Max |
| RN50 | 76.28 | 64.81 | 58.70 | 57.97 | 50.91 | 49.09 | 51.95 | 50.76 | 25.24 | 34.95 |
| RN101 | 72.97 | 72.92 | 63.77 | 62.32 | 50.76 | 51.36 | 46.19 | 46.19 | 19.45 | 06.80 |
| RN50x64 | 71.26 | 47.42 | 57.97 | 57.97 | 58.61 | 49.24 | 57.02 | 57.02 | 07.77 | 08.74 |
| ViT-B/32 | 72.90 | 72.88 | 57.25 | 57.97 | 51.51 | 49.09 | 56.35 | 56.35 | 17.48 | 19.42 |
| ViT-L/14 | 73.36 | 73.07 | 59.42 | 60.14 | 68.13 | 49.40 | 62.61 | 59.90 | 20.38 | 12.62 |
| Best Acc$_{DP}$ | **76.28** | 73.07 | **63.77** | 62.32 | **68.13** | 51.36 | **62.61** | 59.90 | 25.24 | **34.95** |
| RN50 | 73.00 | 76.40 | 59.42 | 54.35 | 53.17 | 42.75 | 55.84 | 51.61 | 23.30 | 29.13 |
| RN101 | 72.95 | 72.97 | 59.42 | 59.42 | 50.76 | 55.74 | 51.27 | 53.64 | 13.60 | 14.56 |
| RN50x64 | 75.03 | 72.69 | 57.97 | 57.97 | 49.85 | 49.40 | 57.02 | 57.02 | 33.01 | 30.01 |
| ViT-B/32 | 72.92 | 72.95 | 46.38 | 42.03 | 50.76 | 50.76 | 56.68 | 56.85 | 14.56 | 14.56 |
| ViT-L/14 | 72.97 | 72.95 | 63.77 | 60.14 | 64.65 | 54.38 | 58.04 | 57.87 | 18.45 | 20.39 |
| Best Acc$_{BP}$ | 75.03 | **76.40** | **63.77** | 60.14 | **64.65** | 55.74 | **58.04** | 57.87 | **33.01** | 30.01 |

*Table 5.* Accuracy (%) of our method and supervised learning methods on the four datasets we use in our work. Note that the accuracy of our method is evaluated on the whole dataset, while results of supervised approaches are calculated on test sets split by the authors. ˜ denotes approximate values, indicating that the accuracies are slightly different from those reported above.

|  | Montgomery | Shenzhen | Pneumonia | IDRID |
|---|---|---|---|---|
| Supervised learning (Baseline) | 62.58 (Sirshar et al., 2021) | 65.70 (Sirshar et al., 2021) | 61.19 (Szepesi & Szilágyi, 2022) | 52.28 (Wu et al., 2020) |
| Supervised learning (SOTA) | 88.40 (Sirshar et al., 2021) | 81.11 (Sirshar et al., 2021) | 95.67 (Szepesi & Szilágyi, 2022) | 56.19 (Wu et al., 2020) |
| Best Acc (Ours) | ˜63.77 | ˜68.13 | ˜76.28 | ˜34.45 |

