# OpenReview forum: "A ChatGPT Aided Explainable Framework for Zero-Shot Medical Image Diagnosis"
_ICML.cc/2023/Workshop/IMLH — IMLH 2023 Poster_

### Official Review · Reviewer_bFiP · 2023-06-12
**A review for A review for “A ChatGPT Aided Explainable Framework for Zero-Shot Medical Image Diagnosis.”**

**Rating:** 6
**Confidence:** 4

**Review:**

This paper proposes an image diagnostic supporting system that leverages the Chatgpt. The framework requires a pre-trained vision-language model, e.g., CLIP, and designs a task-specific prompt as inputs to the Chatgpt, with the output as text descriptions to the CLIP to obtain the disease label embedding. The final predicted label is based on the associate score of the input image and tags embedding.

The proposed framework is easy to follow and intuitive. As a further improvement: 1)the current prompt design is not that interesting, which can be further improved. According to the paper, the performance of the IDRID dataset is not satisfactory yet, which might provide insight into how to improve the current framework.

---

### Official Review · Reviewer_rF2q · 2023-06-17
**This paper proposes a framework for explainable zero-shot medical image classification. The approach leverages Language and Vision models (LLMs) to generate additional cues and knowledge, enhancing accuracy and explainability in medical diagnosis.**

**Rating:** 7
**Confidence:** 4

**Review:**

(+) The proposed framework and the main intuition behind it are somewhat interesting and novel.

(+) The paper's writing level is good.

(+) The experimental section is organized satisfactorily to extensively evaluate the applicability of the proposed method across five datasets.

(-) Despite the authors' efforts to list relevant studies to their approach, the pros and cons of existing studies and approaches are less clear when compared to the proposed method. I would recommend the authors to provide a clearer trajectory in the related work section.

(.) I would recommend the authors to improve the figure captions to enhance clarity and understanding.

(.) I would advise the authors to revise the paper's abstract, introduction, and conclusion to make the paper's goal and messages clearer. For instance, the abstract section may need to be revised, as it is too long and can be difficult to follow at times.

(.) I would suggest that the authors consider including a technical discussion in the paper. A more intuitive and technical justification will increase the paper's credibility and confidence.

---

### Official Review · Reviewer_65zV · 2023-06-18
**Interesting and promising chatgpt-integrated framework in medical tasks**

**Rating:** 7
**Confidence:** 4

**Review:**

This paper present a framework for zero-shot medical image diagnosis using CLIP and ChatGPT. They provide explainable and accurate diagnoses by leveraging pretrained vision-language models and generating additional cues and knowledge.


- They combine the power of CLIP and ChatGPT to enhance explanation in medical tasks, which is interesting and promising.
-  The proposed method demonstrates significant improvements in diagnostic accuracy.
- The paper reads well and the authors also clearly discuss the limitations of their approach.

Some questions:
- What if ChatGPT delivers some inaccurate information in medical queries? Testing different prompts and showcasing the outcomes in the paper would provide insights into this aspect.
- It could be better to discuss the differences and their superiority over other chatgpt-based models in similar tasks.

---

### Meta-Review · Area_Chair_t2uK · 2023-06-20

**Recommendation:** Accept (Poster)
**Confidence:** 5

**Metareview:**

The authors designed a ChatGPT-aided explainable method for medical image diagnosis. The paper proposes to use a CLIP-based zero-shot medical image classification in addition to ChatGPT's assistance. The paper received positive ratings from all reviewers, who believe it holds significant potential for contributing to the community. I suggest the authors can further improve the paper regarding the reviewer's comments.

---

### Decision · Program_Chairs · 2023-06-20

Accept (Poster)